# Semi-Supervised Video Salient Object Detection Based on Uncertainty-Guided Pseudo Labels

**Yongri Piao**[1,*], **Chenyang Lu**[1,*], **Miao Zhang**[1(✉)], **Huchuan Lu**[1,2]

[1]Dalian University of Technology, China   [2]Pengcheng Lab, Shenzhen, China

`yrpiao@dlut.edu.cn`, `luchenyang0724@mail.dlut.edu.cn`,
`{miaozhang, lhchuan}@dlut.edu.cn`

## Abstract

Semi-Supervised Video Salient Object Detection (SS-VSOD) is challenging because of the lack of temporal information caused by sparse annotations in video sequences. Most works address this problem by generating pseudo labels for unlabeled data. However, error-prone pseudo labels negatively affect the VOSD model. Therefore, a deeper insight into pseudo labels should be developed. In this work, we aim to explore 1) how to utilize the incorrect predictions in pseudo labels to guide the network to generate more robust pseudo labels and 2) how to further screen out the noise that still exists in the improved pseudo labels. To this end, we propose an Uncertainty-Guided Pseudo Label Generator (UGPLG), which makes full use of inter-frame information to ensure the temporal consistency of the pseudo-labels and improves the robustness of the pseudo labels by strengthening the learning of difficult scenarios. Furthermore, we also introduce adversarial learning to address the noise problems in pseudo labels, guaranteeing the positive guidance of pseudo labels during model training. Experimental results demonstrate that our methods outperform existing semi-supervised method and partial fully-supervised methods across five public benchmarks of DAVIS, FBMS, MCL, ViSal, and SegTrack-V2. *Code and dataset are available at* `https://github.com/Lanezzz/UGPL`.

## 1   Introduction

Video Salient Object Detection (VSOD) aims to locate and segment the objects people are most interested in in the video consequence. With the increasing demand for video data processing, research on VSOD has received increased attention. As a fundamental technique in computer vision, many video-related applications adopt it as preprocessing to allocate more attention to salient regions, such as video tracking [55] , video object segmentation [37], video action recognition [31], video captioning [41].

Compared with still-image-based SOD tasks, VSOD is more challenging because the prediction of salient objects in video is heavily dependent on temporal dynamics. In recent years, deep-learning-based VSOD has achieved significant progress with the development of CNN. However, it is usually expensive and difficult to label temporally consistent pixel-level annotations for videos. To reduce the heavy efforts of labeling segmentation ground truth, numerous pseudo-label based semi-supervised video methods [58, 3, 12] are proposed. These works try to use limited, sparsely annotated ground-truth labels to generate labels for unlabeled data through self-training or label propagation, to make up for the lack of temporal information between sparsely annotated frames. Because video data has much redundant information (*et al.*, 24 fps in the DAVIS [43] dataset), we believe using a small amount of ground truth and large amounts of pseudo labels to supervise the model's training tends to be more efficient for VSOD.

---

*means equal contributions. Miao Zhang[✉] is the corresponding author.

36th Conference on Neural Information Processing Systems (NeurIPS 2022).

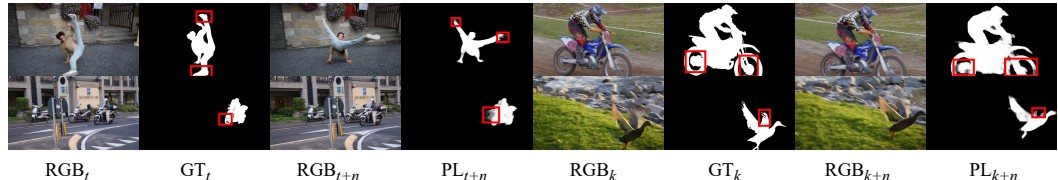

| RGB$_t$ | GT$_t$ | RGB$_{t+n}$ | PL$_{t+n}$ | RGB$_k$ | GT$_k$ | RGB$_{k+n}$ | PL$_{k+n}$ |

Figure 1: PL$_{t+n}$ and PL$_{k+n}$ are the pseudo labels generated by the pseudo label generation network in RCR [58] for RGB$_{t+n}$ and RGB$_{k+n}$, respectively. The areas corresponding to the red box is error-prone. These four scenarios correspond to deformed objects, details of complex objects, interference of similar objects, and blurred boundaries caused by fast motion.

Yan *et al.* [58] try to address semi-supervised VSOD by proposing a teacher network trained with the sparsely sampled annotations (sample every n frames, n > 1) to generate pseudo labels for the unlabeled data, which are involved in the subsequent training of the video saliency model with manually annotated labels. Despite their great performance, further refinements for pseudo labels should be concerned. We observe that pseudo labels tend to have obvious errors regarding difficult scenarios, such as deformed objects, interference from similar objects, and complex contours(shown in Fig. 1). Error-prone pseudo labels are usually detrimental to final detection. Inspired by these observations, we think two key issues need to be considered: 1) How do we use the incorrect information in pseudo labels to guide the pseudo labels generation network to generate more reliable predictions in difficult scenarios. 2) How do we suppress the noise that still exists in pseudo labels for further improving the video saliency model.

In this paper, we strive to overcome difficulties toward accurate semi-supervised VSOD. The primary challenge toward this goal is to generate high-quality pseudo labels and avoid the interference of the noise that still exists in the pseudo labels. The key aspect in the success of our method is to utilize the erroneous prediction in pseudo labels to strengthen the robustness of the pseudo label generator and formulate noise suppression as an adversarial learning problem. Concretely, our contributions are threefold:

- We propose a pseudo label generator that is equipped with an Uncertainty-Aware Dual Decoder Module(UADDM). It makes full use of the inter-frame information to locate the salient objects in unlabeled frames, ensuring the temporal consistency of pseudo labels. To our best knowledge, we first utilize the erroneous prediction in pseudo-labels to strengthen the robustness of the pseudo-label generator by proposing UADDM in semi-supervised VSOD.
- We introduce adversarial learning to learn the distribution of the sparsely labeled training set, and the regions that do not conform to the ground truth distribution in pseudo labels can be filtered out by the generative adversarial network.
- We conduct extensive experiments on 5 widely-used datasets and demonstrate that our method outperforms existing semi-supervised VSOD method and partial fully-supervised methods.

## 2   Related Work

### 2.1   Salient Object Detection

During the past decades, large amounts of conventional methods have been developed for SOD. Early works [24, 2, 59] mainly rely on intrinsic cues (*e.g.*, color and texture) to extract saliency features. However, because low-level features cannot capture rich contextual semantic information, the application scenarios of these traditional methods are very limited. Later on, with the development of deep learning techniques, deep-learning-based methods (*et al.*, [15, 6, 56, 45, 9, 22, 63, 23, 21]) are dominant in this field, which can be divided into integration-based models and edge-based models. Integration-based models aim to aggregate multi-scale features to leverage context information of different levels. Hou *et al.* [15] introduce short connections by linking the deep layers towards shallower ones to integrate features of different layers. Wu *et al.* [56] aggregate partial high-level features to generate a coarse attention map to guide the network to output more accurate maps.

Edge-based models attempt to generate prediction maps with clear boundaries by making better use of edge information. Qin *et al.* [45] Propose a hybrid loss that fuses BCE, SSIM, and IoU to supervise the training process on pixel-level, patch-level, and map-level. Feng *et al.* [9] introduces a boundary-enhanced loss as an assistant to learning exquisite object contours which saves the post-processing operations to refine the boundaries. Wei *et al.* [54] design pixel position aware loss to assign higher weights to edge positions which can help the network focus more on boundary regions. More detailed descriptions of research in SOD field can be approached in related literatures [50, 65, 19, 30, 60, 44, 61].

## 2.2 Video Salient Object Detection

Existing VSOD methods can be divided into two categories: conventional models and deep learning models. Traditional methods rely on people's prior knowledge to extract hand-crafted features, such as color-contrast [34], background prior [57] and morphology cues [46]. Because low-level features have limited representational ability, these conventional methods are usually only suitable for specific scenarios. With the development of CNN, deep-learning-based VOSD models have recently achieved great success. Fan *et al.* [8] present a baseline model with a saliency-shift-aware convLSTM module to progressively integrate temporal information. Li *et al.* [29] propose a dual-stream network to enhance appearance features with motion features or motion saliency. Gu *et al.* [13] design a 3D pyramid constrained self-attention module to capture local motion cues of salient objects. Liu *et al.* [62] proposes a dynamic context-sensitive filtering module to dynamically generate context-sensitive convolution kernels and introduce a bidirectional fusion strategy to fuse spatial and temporal features. However, the remarkable performance achieved by these methods relies on densely annotated video datasets which cost considerable expense and time. Researchers explore addressing VSOD by using weak supervision or semi-supervision to relieve the burden of handcrafted labeling. Zhao *et al.* [64] designs multiple losses from the perspectives of boundary, structure, and front-background similarity to learn SOD by using fixation guided scribble annotations. Yan *et al.* [58] proposes to utilize a self-training method to generate pseudo labels to ease the effort of acquiring high-quality manually annotated data.

## 2.3 Semi-supervised Semantic Segmentation

Existing semantic segmentation methods are also being actively explored to reduce the utilization of large amounts of annotated data. There are two mainstream methods in Semi-supervised Image Semantic Segmentation. The first is consistency regularization which applies different perturbation to the same image and requires their predictions or intermediate features to be consistent [10, 40, 25]. The second method is based on self-training [32, 38, 16]. A teacher network is firstly trained with a small number of ground truth labels and then the network is used to generate pseudo-labels for unlabeled data. Finally, a student model is jointly trained with ground truth labels and pseudo-labels.

Due to the lack of strong data augmentation for video data, it is difficult to apply the consistency regularization to video-related tasks. Current Semi-supervised Video Semantic Segmentation mainly relys on self-training. Chen *et al.* [3] train a teacher network to generate pseudo-labels for unlabeled data and then train the student network with human-annotated labels and pseudo-labels iteratively. Ganeshan *et al.* [12] combines optical flow and the prediction of models to generate a coarse pseudo-label, followed by a refine network to achieve better pseudo labels.

# 3 Our methods

## 3.1 Overview

To avoid low-quality pseudo labels that negatively affect VSOD, our method aims to progressively enhance the quality of pseudo labels and strictly filter out the noise in pseudo labels: (shown in Fig. 2)) We propose an Uncertainty-Guided Pseudo Label Generator (UGPLG) which uses inter-frame information to generate consecutive pseudo labels and enhances the learning of difficult scenarios to further improve the robustness of pseudo labels. 2) We introduce adversarial learning for noise suppression (NS-GAN) to screen out the regions in pseudo labels that do not conform to the distribution of ground truth, making pseudo labels more reliable.

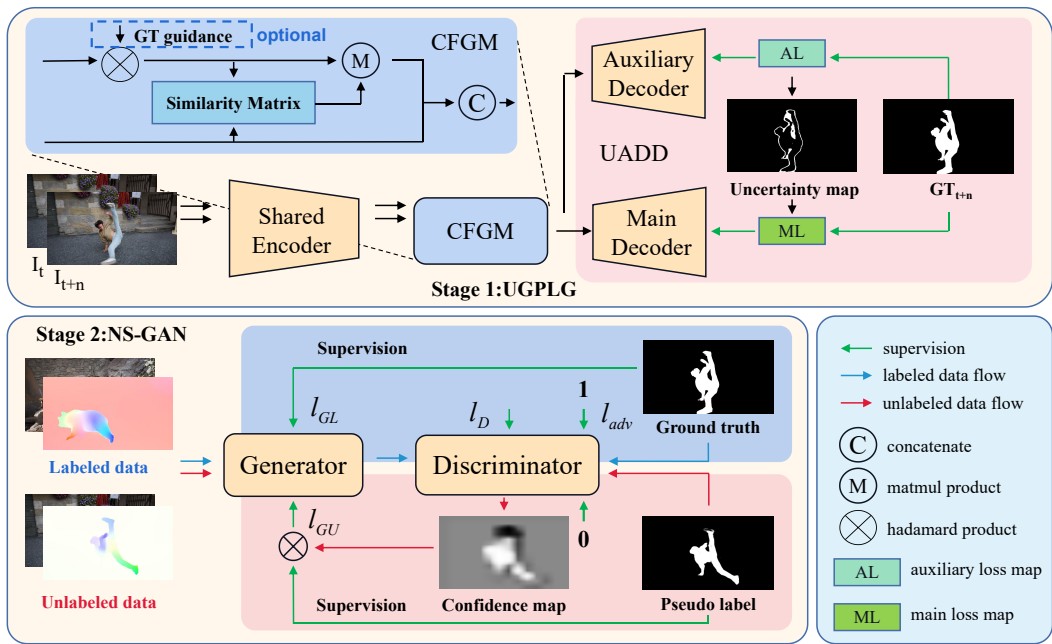

Figure 2: The overall process of our method.

The proposed UGPLG (shown in **Stage 1** of Fig. 2) follows the encoder-decoder architecture. The sharing encoder is based on ResNet50 [14] with ASPP [4]. It consists of a Cross-Frame Global Matching Module (CFGMM) and an Uncertainty-Aware Dual Decoder Module (UADDM). CFGMM aims to refer to the saliency cues in the same video to guarantee the temporal consistency of pseudo labels. And UADDM aims to enhance the learning of uncertain regions in pseudo labels to improve the robustness of pseudo labels. Equipped with the two modules, UGPLG can generate high-quality pseudo labels for complex scenerios.

The proposed NS-GAN (shown in **Stage 2** of Fig. 2) uses a video saliency model as the generator. In the process of adversarial learning, the discriminator can accurately distinguish between the ground truths and the predictions of the generator. Thus by using the discriminator to filter the noise for the pseudo labels, the training process of the generator is free from the negative effects of noise.

### 3.2 Uncertainty-Guided Pseudo Label Generator

**Cross-Frame Global Matching.** Generating pseudo labels for unlabeled video data is very dependent on temporal information because it contains motion cues and location information of the salient objects. The generated pseudo labels are temporally inconsistent if temporal information is not fully utilized. A straightforward utilization of temporal information [11] is computing the optical flow to warp labels to the unlabeled frames. However, optical flow generated from [17] [48] usually cannot accurately predict the geometric relationship between two frames within a long time interval. Therefore, to model the long-term temporal relationship, we select inter-frame global matching which can ignore time intervals to build a bridge for the saliency information propagation between labeled frames and unlabeled frames.

The whole propagation process is divided into two steps. In the first step, shown as **Stage 1** in Fig. 2, given two video frames $I_t$ and frame $I_{t+n}$ in the same video, $F_t \in \mathbb{R}^{C \times (HW)}$ and $F_{t+n} \in \mathbb{R}^{C \times (HW)}$ represent the corresponding features from the sharing encoder. We aim to compute the similarity matrix $S$ between $F_t$ and $F_{t+n}$ by leveraging the non-local [53] mechanism:

$$S = (flat(F_t * L_t))^{\mathsf{T}} W flat(F_{t+n}) \in \mathbb{R}^{(HW) \times (HW)}, \tag{1}$$

where $W \in \mathbb{R}^{C \times C}$ means a weight matrix and $flat$ means the operation of flatten. Then we normalize $S$ in a column-wise way with a softmax function:

$$S^c = softmax(S) \in \mathbb{R}^{(HW) \times (HW)}. \tag{2}$$

Each column of the matrix $S$ represents the relevance between a position on the unlabeled frame and each position on the salient regions of the labeled frame. In the second step, we aim to find out the features in $F_{t+n}$ that are most correlated to salient objects in $F_t$. The transposed $F_t$ is multiplied by the similarity matrix $S$, and we can get the warped feature $\widetilde{F}_{t+n}$ propagated from $F_t$,

$$\widetilde{F}_{t+n} = F_t^{\mathsf{T}} S^c \in \mathbb{R}^{C \times (HW)}. \tag{3}$$

Each element of $\widetilde{F}_{t+n}$ represents the sum of the similarity of all positions on $F_t$ and current position on $F_{t+n}$. By concatenating $\widetilde{F}_{t+n}$ and $F_{t+n}$, we can achieve $F_{fuse}$ which not only retains the spatial information of current frame but also integrates the prior information about salient objects of other frames in the same video. In this way, our UGPLG can generate temporally consistent pseudo labels.

$$F_{fuse} = [F_{t+n}, \widetilde{F}_{t+n}] \in \mathbb{R}^{2C \times (HW)}, \tag{4}$$

where $[\,\cdot\,,\,\cdot\,]$ means concatenation operation.

**Uncertainty-aware Dual Decoder.** Since conventional pseudo label generators are difficult to make accurate predictions in difficult scenarios, to achieve robust pseudo labels, we design an Uncertainty-Aware Dual Decoder Module (UADDM) to detect and utilize the uncertain regions in pseudo labels to improve the performance of UGPLG. During training, when we feed the temporally enhanced feature $F_{fuse}$ to the UADDM, the auxiliary decoder makes a coarse prediction for $I_{t+n}$ and it is supervised by $L_{t+n}$:

$$l_{aux} = l_{bce}(Aux(F_{fuse}), L_{t+n}) \in \mathbb{R}^{H \times W}, \tag{5}$$

where $Aux(\cdot)$ means the auxiliary decoder and $L_{t+n}$ means the label of $I_{t+n}$. Due to the lack of a superior training strategy, the prediction map of the auxiliary decoder often has obvious errors regarding difficult scenarios. When a certain position of $l_{aux}$ is greater than a fixed value, we consider the auxiliary decoder's prediction for that position to be uncertain. Since the structure of the main decoder is exactly the same as that of the auxiliary decoder, the prediction error of the auxiliary decoder in difficult scenarios has a great probability to occur on the main decoder. We assign a higher weight to the uncertain regions the auxiliary decoder predicts to guide the main decoder to train the network to generate more robust pseudo-labels.

$$weight = 1 + \alpha * \mathbb{I}(l_{aux} > T_{loss}) \in \mathbb{R}^{H \times W}, \tag{6}$$

$$l_{main} = weight * (l_{bce}(Main(F_{fuse}), L_{t+n}) + l_{iou}(Main(F_{fuse}), L_{t+n})), \tag{7}$$

where $\mathbb{I}(\cdot)$ is the indicator function, and $Main(\cdot)$ means main decoder. $T_{loss}$ is a fixed threshold which is empirically set as 0.3, and $\alpha$ is empirically set as 5. Thus, the pseudo labels generated by UGPLG can be more robust. As training goes on, the ability of the auxiliary decoder is gradually improved, the number of erroneous predictions is gradually reduced, and the uncertain areas also gradually shrink. Therefore the auxiliary decoder can better guide the main decoder to pay more attention to the real complex scenarios. It is noted that the auxiliary decoder is used only in the training stage. Apart from the standard BCELoss, we also introduce IoU loss to enable the model to focus on global structure. $l_{bce}$ and $l_{iou}$ are defined as:

$$l_{bce} = -\sum_{x=1}^{HW}\sum_{y=1}^{HW}(L(x,y)log(P(x,y)) + (1 - L(x,y))log(1 - P(x,y))), \tag{8}$$

$$l_{iou} = 1 - \frac{\sum_{x=1}^{HW}\sum_{y=1}^{HW} L(x,y)P(x,y)}{\sum_{x=1}^{HW}\sum_{y=1}^{HW} L(x,y) + P(x,y) - L(x,y)P(x,y)}, \tag{9}$$

where $(x, y)$ means the location of the pixel, $H$, $W$ means the height and width, $P(\cdot)$ means the prediction of the model, and $L(\cdot)$ means ground truth.

### 3.3 Adversarial Learning for noise suppression

**Generative Adversarial Network.** To further improve the reliability of pseudo labels, we propose a GAN-based noise suppression mechanism to filter out the regions that do not conform to the distribution of ground truth. The generator is trained to confuse the discriminator so that the distribution of the generator can be closer to the ground truth distribution. Correspondingly, in the process of adversarial learning, the ability of the discriminator also becomes stronger. In that case, the discriminator can accurately identify the noise that still exists in the pseudo labels. By suppressing the noise, we can ensure that pseudo labels are always providing positive guidance to the generator.

The detailed structure of the generator is shown in Fig. 3, which is a classic two-stream network. The upper symmetrical branch is the RGB branch which learns information such as color and outline, and the lower branch is the optical flow branch which learns information such as position and semantics. The two symmetrical branches perform information fusion through a well-designed cross-modality information integration module CRM [20]. Since the information of the two branches is

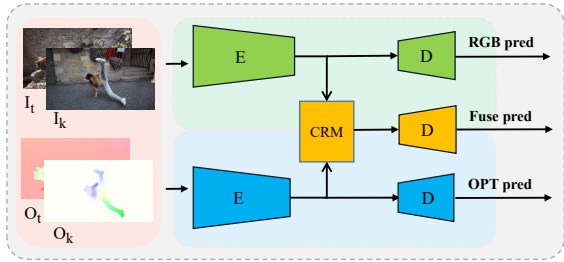

Figure 3: The overall architecture of our video saliency model.

complementary, the fused features of the aggregation branch retain both local detailed information and global structural information. The generator takes both labeled and unlabeled data as input and outputs the predictions of the three branches. The encoder of each symmetrical branch is based on ResNet50 [14] with ASPP [4]. The decoder of the symmetrical branch consists of three convolutional layers, and the decoder of the aggregation branch has one more than that of the symmetrical branch. Similar to [16], our discriminator network is based on FCN [35]. It takes the prediction maps of the saliency model or ground truth labels as input and then outputs a probability map of size $H \times W \times 1$.

**Training with labeled data.** The generator and the discriminator are alternately trained on labeled data. For the training of the discriminator. We minimize the loss using:

$$l_D = l_{bce}(D(G_{fuse}(I_l, O_l)), M0) + l_{bce}(D(Y_l), M1), \tag{10}$$

where $G_{fuse}(\cdot)$ is the output of the aggregation branch in the saliency model, $M0$ is an all-zero matrix and $M1$ is an all-one matrix. $I_l$, $O_l$, and $Y_l$ are the image, optical flow, and ground truth of the labeled frame, respectively. When the input is from ground truth, the discriminator output an all-one map, and when the input comes from the prediction map of the generator, the discriminator output an all-zero map.

To enable the generator to generate prediction maps with the similar distribution as ground truth, $G_{rgb}(\cdot)$, $G_{opt}(\cdot)$, $G_{fuse}(\cdot)$ are all supervised by standard BCEloss, and $G_{fuse}(\cdot)$ is additionally supervised by the adversarial loss $l_{adv}$. The total loss of the generator supervised by labeled data $l_{GL}$ is defined as:

$$l_{GL} = l_{bce}(G_{rgb}(I_l), Y_l) + \delta(l_{bce}(G_{opt}(O_l), Y_l)) + l_{bce}(G_{fuse}(I_l, O_l), Y_l), \tag{11}$$

Where $\delta$ is a hyperparameter with its value of 0.2, and $G_{rgb}(\cdot)$, $G_{opt}(\cdot)$ mean the prediction map of RGB branch, Optical Flow branch, respectively. $l_{adv}$ is defined as:

$$l_{adv} = -\sum log(D(G_{fuse}(I_l, O_l))). \tag{12}$$

With $l_{adv}$, the generator is trained to confuse the discriminator by maximizing the probability of the confidence map. Then the distribution of the prediction output by the generator gets closer to the distribution of ground truth.

**Training with unlabeled data.** Before using pseudo labels for supervised training, to avoid noise providing wrong guidance to the saliency model, we need to filter out the noise in pseudo labels at first. We feed the pseudo labels to the discriminator and obtain the corresponding confidence maps. Then the confidence map $D(PL_u)$ is directly used as the weight of $l_{GU}$ which is the total loss of the generator supervised by unlabeled data:

$$l_{GU} = D(PL_u) * (l_{bce}(G_{rgb}(I_u), PL_u) + \delta(l_{bce}(G_{opt}(O_u), PL_u)) + l_{bce}(G_{fuse}(I_u, O_u), PL_u)), \tag{13}$$

Where the $\delta$ is the same as in Eq. 11. $I_u$, $O_u$, and $PL_u$ are the image, optical flow, and pseudo label of the unlabeled frame, respectively. Because the weights of the noisy regions are very small, the noisy regions have little effect on the loss $l_{GU}$, thus avoiding the model learning from the noise in the pseudo-labels, ensuring the stability of saliency model training and helping the model to converge better. At last, the total loss of the generator is composed of the three types of loss:

$$l_G = l_{GL} + l_{adv} + l_{GU}. \tag{14}$$

# 4 Experiments

## 4.1 Datasets

To evaluate the performance of our method, we conduct experiments on five widely-used VSOD datasets for fair comparisons. **DAVIS** [43] is the most popular VSOD dataset, with 50 high-quality fully annotated video sequences. The whole dataset is split into 30 sequences (2079 frames) for training and 20 sequences (1376 frames) for testing. **FBMS** [39] includes 59 natural video sequences which contains 13860 frames while only 780 of them are sparsely annotated. The whole dataset is split into 29 sequences (353 frames) for training and 30 sequences (720 frames) for testing. **ViSal** [51] is the first specially collected dataset for VSOD. It contains 17 video sequences with 193 sparsely annotated frames. **MCL** [26] includes 9 video sequences about objects with fast movement. There are total 463 sparsely annotated frames. **SegTrack-V2** [27] is the earliest VOS dataset which contains 13 videos, with a total of 1025 annotated frames.

## 4.2 Experimental Setup

**Evaluation Metrics.** We use three universally-agreed criterions to evaluate our results, *i.e.*, mean absolute error (MAE) [42], max F-measure ($F_\beta^{max}$) [1] and structure-measure ($S_\alpha$) [7].

**Implementation Details.** Our network is implemented on the Pytorch framework with $4 \times$ GTX 1080Ti GPU and it is also adapted to the MindSpore framework of Huawei with an Ascend-910. We choose the train set of DAVIS and FBMS as our training set to train both Uncertainty-Guided Pseudo Label Generator (UGPLG) and Noise-Suppressed Generative Adversarial Network (NS-GAN). **For the training of UGPLG**, the initial learning rate is set as 0.005 and decays 0.1 times every 25 epochs with a batch size of 8. **For the training of NS-GAN**, the initial learning rate is set as 0.015 and decays 0.1 times every 20 epochs. Images are uniformly resized to $448 \times 448$. We adopt an SGD optimizer in which the momentum and weight decay are set to 0.9, 5e-4. **In the pre-train phase for NS-GAN**, we pretrain the RGB branch on DUTS [49] which is a commonly used static-image SOD dataset. The initial learning rate is set as 0.01 and decays 0.1 times every 30 epochs.

Table 1: Quantitative comparisons of $S_\alpha$, $F_\beta^{max}$ and MAE on five widely-used VSOD datasets. The best three results are shown in **boldface**, red, and blue fonts respectively. $*$ means conventional methods. - means no available results. † means semi-supervised methods.

| Methods | DAVIS $S_\alpha \uparrow$ | $F_\beta^{max} \uparrow$ | MAE↓ | FBMS $S_\alpha \uparrow$ | $F_\beta^{max} \uparrow$ | MAE↓ | ViSal $S_\alpha \uparrow$ | $F_\beta^{max} \uparrow$ | MAE↓ | MCL $S_\alpha \uparrow$ | $F_\beta^{max} \uparrow$ | MAE↓ | SegTrack-V2 $S_\alpha \uparrow$ | $F_\beta^{max} \uparrow$ | MAE↓ |
|---|---|---|---|---|---|---|---|---|---|---|---|---|---|---|---|
| SGSP* | 0.692 | 0.655 | 0.138 | 0.661 | 0.630 | 0.172 | 0.706 | 0.677 | 0.165 | 0.670 | 0.605 | 0.102 | 0.681 | 0.673 | 0.124 |
| SCOM* | 0.832 | 0.783 | 0.048 | 0.794 | 0.797 | 0.079 | 0.762 | 0.831 | 0.122 | 0.569 | 0.422 | 0.204 | 0.815 | 0.764 | 0.030 |
| SCNN | 0.783 | 0.714 | 0.064 | 0.794 | 0.762 | 0.095 | 0.847 | 0.831 | 0.071 | 0.730 | 0.628 | 0.054 | - | - | - |
| DLVS | 0.794 | 0.708 | 0.061 | 0.794 | 0.759 | 0.091 | 0.881 | 0.852 | 0.048 | 0.682 | 0.551 | 0.060 | - | - | - |
| FGRN | 0.838 | 0.783 | 0.043 | 0.809 | 0.767 | 0.088 | 0.861 | 0.848 | 0.045 | 0.709 | 0.625 | 0.044 | - | - | - |
| SSAV | 0.893 | 0.861 | 0.028 | 0.879 | 0.865 | 0.040 | 0.943 | 0.939 | 0.020 | 0.819 | 0.773 | 0.026 | 0.851 | 0.801 | 0.023 |
| PCSA | 0.902 | 0.880 | 0.022 | 0.868 | 0.837 | 0.040 | 0.946 | 0.940 | 0.017 | - | - | - | 0.865 | 0.810 | 0.025 |
| FDS | **0.922** | **0.912** | 0.020 | 0.888 | 0.875 | 0.041 | 0.903 | 0.869 | 0.029 | **0.866** | **0.823** | 0.024 | 0.849 | 0.773 | 0.028 |
| DCF | 0.914 | 0.900 | **0.016** | 0.873 | 0.840 | 0.039 | **0.952** | **0.953** | **0.010** | 0.767 | 0.713 | 0.028 | **0.883** | **0.839** | **0.015** |
| RCR† | 0.886 | 0.848 | 0.027 | 0.872 | 0.859 | 0.053 | 0.922 | 0.906 | 0.026 | 0.820 | 0.742 | 0.028 | 0.842 | 0.781 | 0.035 |
| Ours† | 0.914 | 0.900 | 0.019 | **0.899** | **0.892** | **0.027** | 0.930 | 0.926 | 0.019 | 0.860 | 0.822 | **0.018** | 0.844 | 0.778 | 0.027 |

## 4.3 Comparison with State-of-the-Art

As shown in Table. 1, we compare our methods with two conventional video salient object methods (remarked with $*$): SGSP [33]*, SCOM [5]*, and eight deep-learning based methods: SCNN [47], DLVS [52], FGRN [28], SSAV [8], PCSA [13], FDS [18], DCF [62], RCR [58]. To guarantee fair comparisons, we utilize the widely-used evaluation code provided by [8]. **First**, compared to the existing Semi-Supervised VSOD method RCR, our method

Table 2: Quantitative comparisons with the number of training data used by state-of-the-art method DCF. The colummn Total calculates the sum of training video frames. † means semi-supervised methods.

| Model | Video | Total |
|---|---|---|
| DCF | DAVSOD + DAVIS + VOS | 11.5K |
| RCR† | **20%**(DAVIS + FBMS + VOS) | 1406 |
| Ours† | **10%**(DAVIVS) + FBMS | 568 |

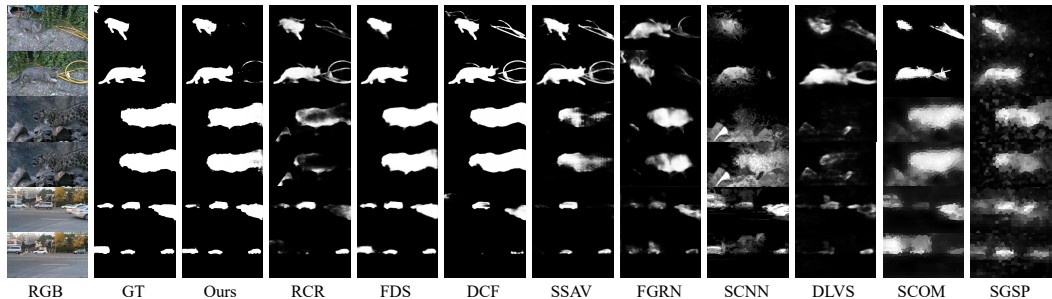

| RGB | GT | Ours | RCR | FDS | DCF | SSAV | FGRN | SCNN | DLVS | SCOM | SGSP |

Figure 4: Qualitative comparisons of state-of-the-art methods. The ground truth (GT) is shown in the foremost column. The prediction map generated by our model in some difficult scenarios can approach or even exceed fully supervised methods.

significantly outperforms them on all datasets.

Specifically, our method improves the MAE by 29.6%, 49.1%, 26.9%, 35.7% on DAVIS, FBMS, ViSal, and SegTrack-V2, respectively, using only 40% of their labeled data (568 versus 1406, shown in Table. 2). **Second**, compared to the fully supervised methods, our method achieves state-of-the-art on FBMS dataset, and achieves 98.7%, 97.2%, 99.9% and 92.7% of the SOTA accuracy on $F_\beta^{max}$ on DAVIS, ViSal, MCL and SegTrack-V2, respectively. Despite our great performance, the number of ground truths we use is only 5% of current state-of-the-art fully-supervised methods. **Finally**, Fig. 4 shows visual comparisons to demonstrate the effectiveness of our proposed method to deal with complex scenarios, such as interference of complex background objects ($1^{st}$ and $2^{nd}$ rows), similar foreground and background ($3^{rd}$ and $4^{th}$ rows) and multiple objects ($5^{rd}$ and $6^{th}$ rows). In these complex cases, it can be observed that our saliency model generalizes better than most VSOD models, which proves that our method can help the network to generate more robust predictions in a variety of complex scenarios.

### 4.4 Ablation Studies

**The Effectiveness of UGPLG.** Uncertainty-Guided Pseudo Label Generator (UGPLG) consists of Cross-Frame Global Matching Module(CFGMM) and Uncertainty-Aware Dual Decoder Module (UADDM), aiming to generate high-quality pseudo labels to guide the training of the saliency model. We perform abundant experiments to validate the effectiveness of each component of UGPLG. As shown in Table. 4, case (1) refers to our baseline model which only retrains the encoder and the main decoder in UGPLG, and the generator in NS-GAN.

**First**, compared to the baseline, case (3) equips with CFGMM additionally. The great performance improvement achieved in case (3) powerfully demonstrates the effectiveness of CFGMM. To further prove the effectivity of CFGMM on introducing temporal information, we propose case (2) by removing the reference to the adjacent ground truth in CFGMM. Compared to case (3), case (2) encounters obvious performance degradation

Table 3: Quantitative comparisons on the quality of the pseudo labels generated for the training set of DAVIS.

| | UGPLG | | | DAVIS$_{train}$ | | |
|------|--------|-------|-------|-----------------|-----------------|------|
| Case | CFGMM-O | CFGMM | UADDM | $S_\alpha \uparrow$ | $F_\beta^{max} \uparrow$ | MAE↓ |
| (1) | - | - | - | 0.937 | 0.927 | 0.016 |
| (2) | ✓ | - | - | 0.943 | 0.931 | 0.015 |
| (3) | | ✓ | - | 0.952 | 0.942 | 0.012 |
| (4) | | ✓ | ✓ | 0.958 | 0.951 | 0.007 |

which demonstrates that our CFGMM can effectively utilize inter-frame information to generate high-quality pseudo labels. **Second**, to prove the effect of UADDM on strengthening the learning of difficult scenarios, we add UADDM (denoted as case (4)) based on case (3). Compared to case (3), the performance improves by 14% and 15% on DAVIS and MCL respectively in terms of MAE. **Moreover**, we directly test the performance of pseudo labels on the DAVIS dataset, shown in Table. 3. Specifically, we train case (1) to case (4) with 10% of the ground truth in DAVIS and then generate pseudo-labels for the remaining 90% of the dataset. The experimental results show that the modules in UGPLG can gradually improve the performance of pseudo labels, finally reaching an impressive performance of 0.007 on MAE. **At last**, we provide the corresponding visualization results in Fig. 5. It can be found that the pseudo labels are getting more consistent temporally. The boundaries of the pseudo labels are getting clearer and the content of salient objects is getting more complete.

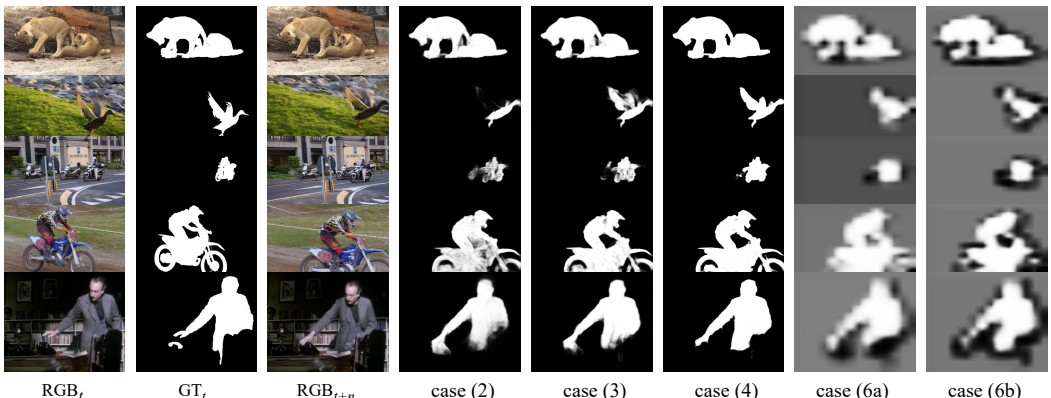

| RGB$_t$ | GT$_t$ | RGB$_{t+n}$ | case (2) | case (3) | case (4) | case (6a) | case (6b) |

Figure 5: Case (2) to case (4) represent the pseudo labels generated by case (2) to case (4). Case (6a) and case (6b) are the confidence maps generated for case (4) by the discriminator of NS-GAN in the early and middle stages of training process, respectively.

Table 4: Quantitative comparisons of the performance of our method on the DAVIS and MCL datasets.

| | UGPLG | | | NS-GAN | | DAVIS | | | MCL | | |
|---|---|---|---|---|---|---|---|---|---|---|---|
| Case | CFGMM-O | CFGMM | UADDM | GAN | NS | MAE↓ | $S_\alpha$ ↑ | $F_\beta^{max}$ ↑ | MAE↓ | $S_\alpha$ ↑ | $F_\beta^{max}$ ↑ |
| (1) | - | - | - | - | - | 0.031 | 0.900 | 0.887 | 0.027 | 0.839 | 0.803 |
| (2) | ✓ | - | - | - | - | 0.030 | 0.902 | 0.887 | 0.027 | 0.842 | 0.807 |
| (3) | - | ✓ | - | - | - | 0.028 | 0.907 | 0.892 | 0.026 | 0.848 | 0.803 |
| (4) | - | ✓ | ✓ | - | - | 0.024 | 0.909 | 0.888 | 0.022 | 0.854 | 0.805 |
| (5) | - | ✓ | ✓ | ✓ | - | **0.019** | 0.910 | 0.894 | 0.019 | **0.860** | 0.815 |
| (6) | - | ✓ | ✓ | ✓ | ✓ | **0.019** | **0.914** | **0.900** | **0.018** | **0.860** | **0.822** |

**The Effectiveness of NS-GAN.** To highlight the importance of the noise suppression mechanism, we conduct a comparative experiment between an ordinary GAN [36] (denoted as case (5)) and NS-GAN (denoted as case (6)), shown in Table. 4. It can be observed that under the condition that case (5) has achieved a great performance, our NS-GAN still creates a further performance improvement which demonstrates the effectiveness of further suppressing the noise of high-quality pseudo labels. Case (6a) and case (6b) in Fig. 5 show the confidence maps generated by the discriminator at different training stages for case (4). As the training process goes on, the discriminator can accurately identify noise areas, ensuring that pseudo labels always provide positive guidance to the saliency model.

**The Effectiveness of Multiple Output Streams.** To explore the importance of all output streams in the generator, we compare the performance differences of supervised training under four output configurations, shown in Table. 5. F, R, O are fused features branch, rgb features branch, and flow features branch, corresponding to Fuse pred, RGB pred, OPT pred in Fig. 3 of the paper, respectively. The training set is 20% of annotated frames. As we can see that the outputs we employ for the generator help the model achieve the best performance.

Table 5: The utility of all output streams.

| | DAVIS | | | MCL | | |
|---|---|---|---|---|---|---|
| Cfg | $S_\alpha$ ↑ | $F_\beta^{max}$ ↑ | MAE↓ | $S_\alpha$ ↑ | $F_\beta^{max}$ ↑ | MAE↓ |
| F | 0.867 | 0.834 | 0.043 | 0.821 | 0.748 | 0.038 |
| RF | 0.874 | 0.843 | 0.039 | 0.830 | 0.772 | 0.027 |
| OF | 0.882 | **0.867** | 0.036 | 0.848 | **0.793** | 0.029 |
| ROF | **0.889** | 0.864 | **0.033** | **0.849** | 0.788 | **0.026** |

**Different Ratios of Ground Truth to Pseudo Labels.** We set up multiple comparative experiments to verify the effectiveness of our method on different ratios of ground truths (GT) to pseudo labels (PL). Table. 6 shows that in all three configurations, our method using 50% pseudo labels already achieves the results that are comparable to the fully-supervised VSOD methods, which fully demonstrates the superiority of our method. Finally, since the training time with 90% pseudo-labels is more than double that with 50% pseudo labels and considering the trade-off between performance and training time, we chose the configuration $M$ using 50% pseudo labels as our final configuration.

Table 6: Quantitative comparisons on different ratios of GT to PL. The best three results are shown in **boldface**, red, and blue fonts, respectively. The configuration $S$ contains 5% GT of DAVIS and 50% GT of FBMS. The configuration $M$ contains 10% GT of DAVIS and 100% GT of FBMS. The configuration $L$ contains 30% GT of DAVIS and 100% GT of FBMS. We generate pseudo labels for all unused data in DAVIS and FBMS. All proportional experiments are divided according to the proportion of the pseudo labels for training. All experiments involving pseudo labels are equipped with noise suppression strategy. GT means ground truth and PL means pseudo labels.

| Dataset | Metrics | $S$ | | | | $M$ | | | | $L$ | | | |
|---|---|---|---|---|---|---|---|---|---|---|---|---|---|
| | | 0% | 20% | 50% | 100% | 0% | 20% | 50% | 100% | 0% | 20% | 50% | 100% |
| DAVIS | $S_\alpha \uparrow$ | 0.873 | 0.899 | 0.911 | 0.911 | 0.880 | 0.902 | 0.914 | 0.910 | 0.894 | 0.903 | 0.915 | **0.918** |
| | $F_\beta^{max} \uparrow$ | 0.847 | 0.880 | 0.895 | 0.901 | 0.851 | 0.882 | 0.900 | 0.895 | 0.867 | 0.878 | 0.901 | **0.905** |
| | MAE$\downarrow$ | 0.040 | 0.022 | 0.019 | **0.018** | 0.034 | 0.023 | 0.019 | 0.020 | 0.032 | 0.023 | **0.018** | **0.018** |
| FBMS | $S_\alpha \uparrow$ | 0.870 | 0.885 | 0.893 | **0.897** | 0.887 | 0.889 | 0.889 | 0.892 | 0.885 | 0.891 | 0.896 | 0.891 |
| | $F_\beta^{max} \uparrow$ | 0.860 | 0.880 | 0.890 | 0.893 | 0.873 | 0.879 | 0.892 | 0.894 | 0.877 | 0.881 | **0.901** | 0.893 |
| | MAE$\downarrow$ | 0.054 | 0.034 | 0.030 | 0.028 | 0.043 | 0.032 | **0.027** | 0.031 | 0.041 | 0.032 | 0.031 | 0.032 |

## 5 Conclusion

In this paper, we strive to face the challenge of error-prone pseudo labels are usually detrimental to final detection. We propose an Uncertainty-Guided Pseudo Label Generator, which makes full use of temporal information to generate temporally consistent pseudo labels, and enhances the learning of uncertain regions to improve the robustness of pseudo labels. To screen out the noise that still exists in the pseudo labels, we further propose an adversarial learning strategy for noise suppression to ensure that the pseudo labels can always guide the network positively. Experimental results demonstrate that our method outperforms the existing semi-supervised VSOD method and partial fully-supervised VSOD model while using only 5% labeled data used by the SOTA fully-supervised methods.

**Acknowledgements.** This work was supported by the National Natural Science Foundation of China (62172070 and 61976035), the Central Government Guided Local Science and Technology Development Funds of Liaoning Province (2022JH6/100100028), the Natural Science Foundation of Liaoning Province (2021-MS-123), and CAAI-Huawei MindSpore open fund (CAAIXSJLJJ-2022-014C).

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
