# OpenReview forum: "Semi-Supervised Video Salient Object Detection Based on Uncertainty-Guided Pseudo Labels"
_NeurIPS.cc/2022/Conference — NeurIPS 2022 Accept_

### Official Review · Reviewer_1nrq · 2022-07-09

**Rating:** 5
**Confidence:** 2
**Soundness:** 2 fair
**Presentation:** 3 good
**Contribution:** 3 good

**Summary:**

The paper presents a method for semi-supervised video salient object detection. It adopts Cross-Frame Global Matching Module (CFGMM) to exploit temporal information and uses Uncertainty-Aware Dual Decoder Module (UADDM) to locate uncertain regions with an auxiliary decoder. Adversarial learning is adopted to generate reliable pseudo-labels.

**Questions:**


1.	There are some letters that are not explained clearly.

a)	What does ‘k’ mean in Figure 1, RGB_{k+n}?

b)	What is L_t in Eq.(1)?

2.	In Eq.7, why do you use ‘I_{t+n}’ as the input of the main decoder? Why not F_{t+n} or F_{fuse}?

3.	The loss (the training process) seems a little bit too complicated. The design of the loss functions is not well validated. Do we really need so many loss terms?




**Ethics Review Area:**

["I don’t know"]

**Limitations:**

No. Please include a brief section introducing the limitations of this work (more analysis needed).

**Strengths And Weaknesses:**

Pros:

1.	The topic of semi-supervised video salient object detection is interesting and important.

2.	The proposed method outperforms state-of-the-art semi-supervised video salient object detection methods on multiple datasets.

3.     Good ablation studies.



Cons:

1.	Insufficient literature review, especially some very related research areas, e.g. weakly/semi-supervised segmentation, and uncertainty-aware segmentation/salient detection, semi-supervised image salient object detection.

2.	The novelty of this paper is somewhat limited. 1) Adversarial training has been fully exploited in very related areas, e.g. semantic segmentation [18].

3.	The training process (including the loss terms) is a little bit complicated.

4.	The runtime performance and computational complexity is not reported.

---

> ### Author Response · Authors · 2022-08-02
> **We clarify the contributions of the paper, especially the differences between our NS-GAN and [18]. We also supplement the experiments on the multiple outputs of the video saliency network and the FPS and parameters of our model.**
>
> Dear reviewer, we greatly appreciate the time and effort you have dedicated to providing insightful feedback on ways to strengthen our paper. It is with great pleasure that we make a point-by-point response to all the comments.
>
> **Insufficient literature review.** Thank you for your valuable suggestion. We are sorry that literature review is not more sufficient than intended. As per your advice, we will make up literature about weakly/semi-supervised segmentation, uncertainty-aware segmentation/salient detection and semi-supervised image salient object detection in related work in the final version.
>
>
> **The novelty of our paper.** Thank you for your valuable suggestions. To our best knowledge, we first utilize the erroneous prediction in pseudo-labels to strengthen the robustness of the pseudo-label generator by proposing UADDM in semi-supervised VSOD. We also formulate noise suppression as an adversarial learning problem by filtering out the regions that do not conform the ground truth distribution in pseudo-labels.
>
> Due to the semi-supervised and weakly supervised methods lack ground truth supervision, the pseudo-labels they generate are of poor quality. Therefore, most methods focus on how to improve the quality of pseudo-labels. With this regards, our method is similar in motivation to [18], we both use adversarial learning to suppress noise. However, our method differs from [18] in terms of technical aspects. Their pseudo-labels are highly coupled with the generative adversarial network. The prediction map of their generator is directly used as pseudo-labels, and the pseudo-labels are in turn used to train the generator. In the early stages of training, their pseudo-labels are of poor quality, which negatively affects the training of the model. On the other hand, our pseudo-labels generated by the pseudo-label generator are decoupled from the adversarial learning network. And the temporal consistency and robustness of our pseudo-labels ensures the stability of adversarial learning network training. Thank you for giving us the valuable opportunity to further clarify the innovation of our method and make our paper achieve a better level. We will highlight the novelty of our method in the final version.
>
> **Complicated training process.**
> The complicated training process comes from the generator trained by multiple signals. However, the introduction of multi-modal data is beneficial for adversarial learning strategies, because the RGB information and the motion information can complement each other, being useful for capturing moving salient objects in video sequences. To verity the utility of multi-modal data for the generator, we conduct detailed ablation experiments on the output stream.The detailed ablation experiments are shown in the table below. F, R, O correspond to “Fuse pred”, “RGB pred”, “OPT pred” in Figure 3 of the paper, respectively. The training set is 20% of annotated frames. As we can see that the outputs we employ for the generator help the model achieve the best performance.
> |Dataset|		|DAVIS	|	|	|MCL||
> | :----: | :----: | :----:|:----:|:----:|:----:|:----:|
> |Metrics|	S|	mF|	MAE|	S|	mF|	MAE|
> |F|	0.867|	0.834|	0.043|	0.821|	0.748|	0.038|
> |RF|	0.874|	0.843|	0.039|	0.830|	0.772|	0.027|
> |OF|	0.882|	0.867|	0.036|	0.848|	0.793|	0.029|
> |ROF|	0.889|	0.864|	0.033|	0.849|	0.788|	0.026|
>
> **Confusing letters and Wrong formula.** We are sorry that some letters that are not explained clearly and Eq.(7) does not match Stage 1 in Figure 2. $k$ represents different video sequences and $L_t$ in Eq.(1) is the ground truth of video frame $I_t$. The input of the main decoder should be $F_{fuse}$. Thanks for your careful review and we will make corresponding change in the final version.
>
> **Runtime performance and computational complexity.** Thanks for your
> constructive suggestions. We implement multiple VSOD models in PyTorch, accelerated by an NVIDIA RTX 1080Ti. We classify all methods into ones without using optical flow (RCR, DCFNet) and using optical flow (MGA, FSNet, Ours). We can find that compared with VSOD models using optical flow, our method has a smaller number of parameters and also has a slight advantage in FPS ($\dagger$ means not count flow generation time and $\ast$ means count flow generation time). We also notice that the generation of optical flow is time-consuming, about 0.44s per frame and brings computational burden for achieving real-time application. We will explore the real-time VSOD model without optical flow data in future. Thanks again for your suggestion and giving us the valuable opportunity to strengthen our paper.
>
> |         |	 |w/o      optical|	    |	   |optical|    |   |      |
> | :----: | :----: | :----:|:----:|:----:|:----:|:----:|:----:|:----:|
> |Methods|RCR|DCFNet|MGA$\dagger$|	FSNet$\dagger$|Ours$\dagger$|MGA*|	FSNet*|Ours*|
> |FPS|28|27|34|16|42| 2.11|1.97|2.14|
> |Params|53.8M|71.7M|91.5M|102.3M|83M|91.5M|102.3M| 83M|

---

> > ### Comment · Reviewer_1nrq · 2022-08-08
> > **Thanks for the response**
> >
> > Thanks. I have read the responses and most of my concerns (e.g. presentation and runtime analysis) have been addressed.

---

> > > ### Author Response · Authors · 2022-08-09
> > > **Thanks for the valuable suggestions.**
> > >
> > > Dear Reviewer 1nrq，
> > >
> > > We sincerely appreciate your constructive comments and positive response. We will carefully improve the paper and include the above analyses and results in the revised version.
> > >
> > > Thanks & Regards,
> > > Authors of paper-6161

---

### Official Review · Reviewer_Vb2d · 2022-07-11

**Rating:** 6
**Confidence:** 4
**Soundness:** 3 good
**Presentation:** 3 good
**Contribution:** 3 good

**Summary:**

This paper proposes a pseudo label generation and processing algorithms. The proposed UGPLG generates temporally consistent pseudo labels. The proposed adversarial learning strategy can suppress the noise. The experimental results show that the method can achieve better performance with fewer gts.

**Questions:**


•	Please indicate the dimension of Aux(Ffuse) in Equation 5 and the dimensions of L(.) and P(.) in Equation 8. Why the dimension of laux is H×W. Similarly, why use * in Equation 13, the specific meaning should be explained.
•	From Figure2.Stage1, the input of the main encoder is Ffuse, but in Equation 7, the input of the main encoder is a frame, please explain in more detail.
•	How to get the optical flow data mentioned in the paper?


**Limitations:**

Two main modules are designed in the paper: UGPLG and NS-GAN. In the ablation experiment, only UGPLG is tested individually. But it is also necessary to test the NS-GAN separately.

**Strengths And Weaknesses:**

Strengths
•	Results greatly improve the performance and the algorithm is strongly competitive with current algorithms.
•	From the results of the ablation experiments, the improvements in the article have improved the performance.
•	The semi-supervised method proposed in this article has great application value, and it is very creative to use the discriminator to suppress noise.

Weaknesses
•	The formulas and figures in the paper can be expressed more clearly.

---

> ### Author Response · Authors · 2022-08-02
> **We provide a clearer explanation of the confusing formulas and tables, and carefully analyze the ablation experiments of NS-GAN.**
>
> Dear reviewer, thanks for your valuable feedback on ways to strengthen our paper.
> We are very grateful for the comments and suggestions. Please see below, our
> detailed response to comments.
>
> **Formulas and Figures.** Thank you for your careful review. We are sorry that part of formulas and figures are
> not expressed in a clear way. We will carefully revise our paper to improve the presentation.
>
> (1). The dimension of $Aux$($F_{fuse}$) in Eq.(5) is 1 $\times$ H $\times$ W. Because the ground truth
> is a single-channel binary mask, the output of our model should be consistent with the
> dimension of the ground truth. In addition, when we set the parameter “reduce” of the
> function binary_cross_entropy_with_logits to “false” in code, the loss $l_{aux}$ is calculated element by element to generate a loss map of 1 $\times$ H $\times$ W.
>
> (2). The dimensions of ${L}(\cdot)$ and ${P}(\cdot)$ in Eq.(8) are both H $\times$ W.
>
> (3). In Eq.(13), ${D}(PL_u)$ is the confidence map generated by the discriminator for the
> pseudo-label, whose dimension is H $\times$ W. If the value of a certain position of the
> confidence map is close to 0, it means that the pseudo-label is very unreliable at that
> position. “$\ast$” is used because noisy regions need to be reduced by multiplying a small weight, ensuring the areas with high confidence in pseudo-labels guiding the network training.
>
> (4). We are sorry that Eq.(7) does not match Stage 1 in Figure 2. The input of the
> main decoder should be $F_{fuse}$. Thanks for your careful review and we will make
> corresponding change in the final version.
>
> Thanks for your suggestion and giving us the valuable opportunity to strengthen our
> paper.
>
> **Optical Flow.** There are many excellent deep learning-based optical flow generation
> models, such as FlowNet2 [a], PWCNet [b], RAFT [39], etc. Because the RAFT model is very convenient to install, we finally choose RAFT to generate optical flow.
>
> **Ablation experiments about NS-GAN.** Thank you for your valuable suggestions.
> NS-GAN consists of an adversarial learning strategy and a noise suppression strategy.
> First, it pushes the distribution of the prediction of the generator to the ground-truth
> distribution through adversarial learning strategy. Second, the discriminator is reused
> to detect and suppress the unreliable regions in pseudo-labels. The ablation experiments of the adversarial learning strategy and noise suppression strategy correspond to case (5) and case (6) of Table 4, respectively. We can find that under the condition that case (5) has achieved a great performance, our noise suppression strategy still creates a further performance improvement, demonstrating the effectiveness of further suppressing the noise of high-quality pseudo labels.
>
> [a]. FlowNet 2.0: Evolution of Optical Flow Estimation with Deep Networks. 2016.
>
> [b]. PWC-Net: CNNs for Optical Flow Using Pyramid, Warping, and Cost Volume. 2018.

---

### Official Review · Reviewer_ctPj · 2022-07-12

**Rating:** 5
**Confidence:** 4
**Soundness:** 2 fair
**Presentation:** 2 fair
**Contribution:** 2 fair

**Summary:**

This paper proposes a semi-supervised approach to perform salient object detection in videos using a combination of pseudo-label prediction and adversarial noise reduction training. The authors propose a multi stage training structure where the first focuses on pseudo-label generation using an uncertainty guided approach. The second stage then uses a GAN based structure to improve uncertainty identification and pseudo labels while reducing noise in the overall pseudo label generation. The authors demonstrate the use in multiple datasets and show competitive results for semi-supervised approach.

**Questions:**

Please see weaknesses.

**Limitations:**

The authors have not explicitly discussed limitations of this approach. One would expect certain obvious limitations based on this approach, such as cross frame matching for certain sequences based on motion/content and the training of GAN.


----
Post-rebuttal
The authors have answered the questions from the initial review.

**Strengths And Weaknesses:**

Strengths

- The idea of using temporal information for generating semantically matching pseudo-labels is an interesting approach. While optical flow base warping is generally used for temporal consistency, there are errors and drawbacks in optical flow based warping. However, using a non-local attention based similarity mapping removes those errors for matching features to generate pseudo labels.

- The use of uncertainty in the dual decoder module is also an interesting approach to generate positive bias to learn the pseudo labels. As the pseudo labels are noisy, having different weight bias for each pixel is essential to learn better. Uncertainty has been effectively used for such task in image-based pseudo label learning.

- The authors show the result on one of the biggest VSOD dataset (DAVIS) as well as 4 other dataset and compare with prior fully-supervised and one semi-supervised method. The authors also show in table 5 that the uncertainty based pseudo-label approach can achieve results similar to fully supervised VSOD methods using only 50% pseudo labels.

Weakness

- The overall approach using pseudo labels improves semi and weakly supervised task in general and noise augmentation/suppression works on making network resilient to these pseudo labels. In that regard, the uncertainty aware decoder module as well as the cross frame similarity (global matching) does not have significant contribution on its own. The non-local attention based matching is simple but also already validated approach that is in line with other validated approaches such as optical flow based warping and interpolation. Uncertainty based decoder does have technical merit, so more detailed analysis on that would be a plus for the paper.

- GAN based approach for noise suppression and semi-supervised training is an unconventional take, which is interesting. The issue is with having to train a GAN model with multiple outputs (rgb, optical flow and fused salient features). Is the training process similar for all dataset? While optical flow warping is not considered in the cross frame matching module, the generator is learning to fuse the optical flow information. Have the authors explored utility of all input/output streams in the generator?

- While multi-stage approach is common for semi and weakly supervised approach and the components used here have positive effect on the task, most components are simple use of existing validated approaches with low technical novelty. The uncertainty based dual decoder is more interesting and deserves more exploration.

---

> ### Author Response · Authors · 2022-08-02
> **We clarify the main contributions of the paper, and made a more detailed analysis of UADDM, and finally propose detailed answers to each suggestion or question.**
>
> Dear reviewer, thank you for taking time out of your busy schedule for reviewing our paper. The comments are valuable and constructive, and we will explain your concerns point by point.
>
> **The novelty of overall approach.** Thanks for your insightful suggestions. We greatly appreciate the opportunity to better clarify the novelty of the proposed pseudo-label generator. Most semi-supervised and weakly supervised methods focus on improving the quality of pseudo-labels, due to the lack of full ground truth supervision. To our best knowledge, we first utilize the erroneous prediction in pseudo-labels to strengthen the robustness of the pseudo-label generator by proposing UADDM in semi-supervised VSOD. We also formulate noise suppression as an adversarial learning problem by filtering out the regions that do not conform the ground truth distribution in pseudo-labels.
>
> **The Cross-Frame Global Matching Module (CFGMM).** CFGMM is not considered as our main contribution, belonging to a part of UGPLG. CFGMM aims to locate the salient objects for unlabeled frames, ensuring temporal consistency of pseudo-labels. The first main contribution involving CFGMM of our work is to utilize the uncertain regions in pseudo-labels to improve the robustness of the pseudo-label generator.
>
> **Detail analysis on Uncertainty-Guided Dual Decoder Module (UADDM).** Thanks for your insightful suggestion. we agree with you that more detail analysis on uncertainty can make the contribution of our UADDM conspicuous. According to your suggestion, we will add more detailed description on UADDM. The UADDM is designed to detect and utilize the uncertain regions in pseudo-labels to improve the robustness of the pseudo-label generator. In terms of detection, we first utilize the auxiliary decoder to generate a coarse prediction map for the video frame. Due to the lack of a superior training strategy, the prediction map of the auxiliary decoder often has obvious errors regarding difficult scenarios. Since the structure of the main decoder is exactly the same as that of the auxiliary decoder, the prediction error of the auxiliary decoder in difficult scenarios has a great probability to occur on the main decoder. We directly utilize the uncertain regions the auxiliary decoder predicts to guide the main decoder to train the network to generate more robust pseudo-labels. As training goes on, the ability of the auxiliary decoder is gradually improved, the number of erroneous predictions is gradually reduced, and the uncertain areas also gradually shrink. Therefore the auxiliary decoder can better guide the main decoder to pay more attention to the real complex scenarios. The corresponding visualization results in Case (3) and Case (4) in Figure 5 confirm this claim. Case (4) adds UADDM based on Case (3). As we can see, when encountering difficult scenarios such as distorted bird wings, interference from similar electric vehicles, complex motorcycle contours, and occlusion by chairs, the boundaries of Case (4) are clearer and the salient objects of Case (4) is more complete. This verifies that our UADDM indeed guides the pseudo-label generator to generate more robust pseudo-labels. Thanks again for making our paper to achieve a better level.
>
>
> **Adversarial Learning.** You're right, the same training process is used for all video datasets.
>
> **Optical flow information.** Thanks for your valuable comments. We did not use optical flow to generate pseudo-labels because the existing optical flow models cannot well reflect the motion trajectories of objects in natural images, and using the optical flow to warp ground truth is often accompanied by severe distortion. When training the video saliency model, the purpose of introducing optical flow is to roughly locate the salient objects in the video by using the motion information containing in the optical flow.
>
> **Utility of all input/output streams in the generator.** Per your advice, we conduct detailed ablation experiments on the output stream.The detailed ablation experiments are shown in the table below. F, R, O are fused features branch, RGB features branch, and flow features branch, corresponding to “Fuse pred”, “RGB pred”, “OPT pred” in Figure 3 of the paper, respectively. The training set is 10% of the ground-truth in the DAVIS dataset and 100% of the ground-truth in the FBMS dataset, totally 20% of the annotated frames. As we can see that the outputs we employ for the generator help the model achieve the best performance.
>
> |Dataset|		|DAVIS	|	|	|MCL||
> | :----: | :----: | :----:|:----:|:----:|:----:|:----:|
> |Metrics|	S|	mF|	MAE|	S|	mF|	MAE|
> |F|	0.867|	0.834|	0.043|	0.821|	0.748|	0.038|
> |RF|	0.874|	0.843|	0.039|	0.830|	0.772|	0.027|
> |OF|	0.882|	0.867|	0.036|	0.848|	0.793|	0.029|
> |ROF|	0.889|	0.864|	0.033|	0.849|	0.788|	0.026|

---

> > ### Comment · Reviewer_ctPj · 2022-08-08
> > **Thank you for the response**
> >
> > The authors have addressed the concerns raised in original reviews regarding different components used in the network, their importance, and ablation studies regarding some components. The authors definitely need to highlight the UADDM module in more detail and present that as a major contribution. Regarding the optical flow use, author’s claim of existing OF methods not reflecting objects in natural images properly is not grounded or based on any prior works. In fact, prior works have used OF from models like FlowNet and RAFT along with an occlusion mask to reduce warping errors successfully. Since they mention their objective of using OF is to ‘roughly locate the salient objects’, claiming other OF + warping methods don’t reflect objects in natural images might be contradictory. The utility of all input/output streams is clearer with the provided table.

---

> > > ### Author Response · Authors · 2022-08-09
> > > **Thank you for the valuable feedback**
> > >
> > > Dear reviewer, we greatly appreciate the time and effort you have dedicated to providing insightful feedback. As per your advice, we will highlight the UADDM module in more detail and present that as a major contribution in the final version. On the other hand, we agree with you that optical flow along with an occlusion mask can be used to reduce warping errors successfully[a][b] and we are sorry that our expression was in a less clear way. In the original response “the existing optical flow models cannot well reflect the motion trajectories of objects in natural images, and using the optical flow to warp ground truth is often accompanied by severe distortion”, we originally meant that optical flow is not perfectly accurate, and it is more prone to errors in some special scenarios, such as object occlusion, deformation of salient objects, and object blur caused by high-speed motion, etc. Although the optical flow in some scenarios is not very accurate, it still contains a lot of motion information of salient objects. By efficiently fusing motion information and image information, our model is able to accurately locate and segment salient objects in video sequences and that is why we introduce multi-modal input in the generator. Thanks again for taking the valuable time and providing the insightful comments towards improving our paper and we will provide a more rigorous expression in the final version. Please let us know if you have other questions, and we are happy to address them.
> > >
> > > [a].Efficient Semantic Video Segmentation with Per-frame Inference. 2020.
> > >
> > > [b].Learning blind video temporal consistency. 2018.

---

### Meta-Review · Area_Chair_fMVf · 2022-08-29

**Recommendation:** Accept
**Confidence:** Less certain

**Metareview:**

In this paper the authors propose an approach for semi-supervised salient object detection using a combination of pseudo-label prediction and adversarial training, showing improved results on a number of benchmarks. Some concerns about more detailed analysis of aspects of the approach were raised, but seemed to be mostly addressed in the authors’ response. Some reviewers also expressed concerns about there being sufficient technical contributions, but seemed satisfied enough with the analysis and strong positive results.

**Award:**

No

---

### Decision · Program_Chairs · 2022-09-14

Accept